# Do Global and Local Perform Cooperatively or Adversarially in Heterogeneous Federated Learning?

Huiwen Wu[1] , Shuo Zhang [2,3] *

[1] Research Center for Data Hub and Security, Zhejiang Laboratory, Hangzhou 310000, China;
[2] State Key Laboratory of Mathematical Sciences, Academy of Mathematics and Systems Science, Chinese Academy of Sciences, Beijing 100190, China;
[3] School of Mathematical Sciences, University of Chinese Academy of Sciences, Beijing 100049, China.
`whw@zhejianglab.org, szhang@lsec.cc.ac.cn`

Heterogeneous federated learning (Hetero-FL) is an emerging machine learning framework that enables the training of collaborative models between devices with varying capabilities and data without sharing raw data. In HFL, there are two types of trainer that exhibit distinct behaviors: the Global Trainer (GTr), which prioritizes average performance while lacking fine-grained client insights; the Local Trainer (LTr), which addresses local issues and excels in local data, but struggles with generalization. Thus, it is crucial to combine them, obtaining an admired GTr. Unlike the prevalent personalization strategies that supplement GTr with LTr, our work introduces a novel approach in which GTr and LTr collaborate adversarially. The adversarial performance of the local trainer can unexpectedly enhance the overall performance of GTr in the combined global-local training process. Building on a profound understanding of this adversarial cooperation, we propose an alternating training strategy named Fed A(dversarial) B(ased) (C)ooperation (FedABC), utilizing a "G-L-G-L" framework. LTr increases the global loss, preventing GTr from falling at local minimum points. Our comprehensive experiments show superior accuracy, up to 13.77%, and faster convergence than existing state-of-the-art Hetero-FL methods. We validate the effectiveness and efficiency of our approach in terms of fairness, generalizability, and long-term behavior. Ultimately, our proposed method underscores the design of the training strategy of the Hetero-FL model, emphasizing adversarial cooperation between GTr and LTr in real-world scenarios.

## 1. Introduction

Recently, Federated Learning (FL) has become a widely used technique in machine learning due to the large amount of data stored in various physical locations and the restrictions on data transmission imposed by security and privacy regulations. FL has been successfully applied to a variety of scenarios, such as smart devices [1], cross-silo graph learning [2, 3], and cross-domain recommendation [4, 5], by allowing cooperative training in physically isolated data and clients. This enables the use of distributed data and computing power to achieve a high-quality training effect.

Despite the success of FL, statistical heterogeneity of the data remains a major issue that affects the performance of FL models. For example, `FedAvg` [1], updates a common global model with the synchronization of local gradients every few training steps and the extraction of local information during the two synchronizations, which we denote as the global trainer (`GTr`). The benefits of `GTr` is that it enables collaborative learning using an averaged global model without sharing raw data. However, updating a single common global model does not fit all heterogeneous local clients well. The solely averaged global model can lead to a significant decrease in accuracy, especially when each client has an extremely skew local dataset [6]. On the other hand, the local paradigm (`LTr`) trains each client solely on its own data set, which perfectly captures the local representation, but

---
*Corresponding Author

Second Conference on Parsimony and Learning (CPAL 2025).

ignores the additional information from other clients and the connection between them. Furthermore, `LTr` trained with a limited amount of data would lead to poor generalization ability. Therefore, the key to improving the effect of training is understanding the function of global and local information in FL and how to use them intelligently.

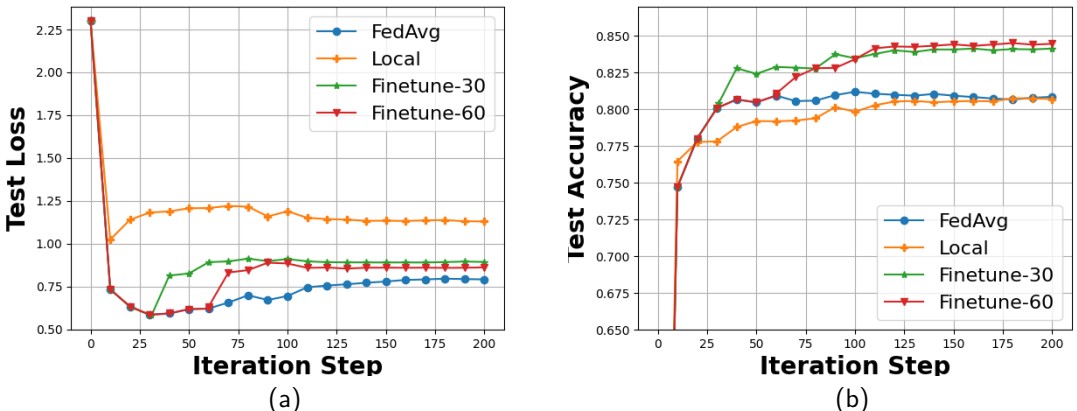

Figure 1: An motivation example with CIFAR10.

Recent research has focused on three primary strategies to address the challenge of statistical heterogeneity. The initial strategy is through fine-tuning, which involves adjusting the pre-trained FL model using a specific local dataset [7–11]. Fine-tuning [9] can transition from global to local setting once, but tends to lose global information over prolonged periods of local adaptation [12]. The second strategy is the reduction of variance, which uses optimization methods to minimize variance between clients [6, 13–17]. The variance in local gradients reflects the diversity of FL clients. Researchers use optimization-driven techniques to mitigate variance during the training process. However, this approach requires periodic computation of the entire gradient using all data samples to reduce data variance, which introduces significant computational complexity. The third approach is the local adapter, which uses local information for fast adaptation of `GTr` models [18–22]. These methods correct the `GTr` by `LTr` information, which fail to discover the interaction between `GTr` and `LTr` thoroughly. These methods tend to find a balance of `LTr` to `GTr`, which does not directly resolve the statistical heterogeneity.

This paper examines two questions related to FL training: Can we achieve better accuracy instead of finding a balance between `GTr` and `LTr`, and what is the relationship between `GTr` and `LTr`, particularly in long-term FL training, cooperative or adversarial? To answer these questions, dedicated research on the training behaviors of `GTr` and `LTr` is necessary. `GTr` aggregates local gradients derived from local loss functions and local data sets in a weighted average fashion, but may overlook fine-grained information specific to local data sets. `LTr` are well-suited to local datasets, but do not generalize to global data sets. It is necessary to combine the advantages of both `GTr` and `LTr`.

A natural solution is fine-tuning the `GTr` pretrained model with `LTr`. Due to the game between `GTr` and `LTr`, the fine-tuning model behaves between `GTr` and `LTr` in test loss while jumping to a high test accuracy (Figure 1). Inspired by the concepts of multiscale and multilevel in signal representation and the resolution of partial differential equations [23–28], this article presents the development of an **A**dversary-**B**ased **C**ooperation (ABC) training approach in FL (`FedABC`). This involves a systematic alternation between `GTr` and `LTr` throughout the FL training process, with stages of alternation predefined. `GTr` and `LTr` optimizers can be viewed as the two scale solvers for the objective function naturally. We will especially focus on the adversarial training strategy that can actually help organize the cooperation between `GTr` and `LTr` optimizers. The adversarial relation of `GTr` and `LTr` is they have different learning directions. Both `GTr` and `LTr` solve optimization problems with different data scales. `GTr` solves for a union of all clients data while `LTr` solves for

its own data. The cooperative relationship lies in the fact that the information grasped by `GTr` and `LTr` is complement. `GTr` tends to grasp the common information among clients while `LTr` focuses on the individual fine-grained information. The main design logic is to alternatively use `GTr` and `LTr` to gain both advantages. Concretely, when the `LTr` arrives the plateau, meaning that the fine-grained information missed by the `GTr` tend to relatively small, we return to run `GTr`. And when `GTr` arrives the plateau, we switch to `LTr`. The alternating between `GTr` and `LTr` trainers tend to enjoy fast convergence in both global and local optimization.

We evaluate the proposed `FedABC` on four FL benchmarks compared to eight SOTA methods in a heterogeneous FL setting. We show that `FedABC` achieves fast convergence and high accuracy compared to all candidate SOTA methods. Specifically, `FedABC` improves test accuracy up to 7.47 % and 11.01% average compared to `GTr` and `LTr` respectively, and much fewer convergence steps to achieve a good accuracy threshold compared to personalized FL frames.

In summary, our main contributions are summarized as follows:

- We observe a phenomenon that an alternating `GTr` and `LTr` strategy results in oscillation in model performance, and this oscillation further helps the model to struggle to a better accuracy.

- Our innovation, called `FedABC`, involves alternating between `GTr` and `LTr` in hetero-FL. This approach leverages complementary information from both `GTr` and `LTr` to achieve fast convergence and better accuracy.

- In practice, we have devised two different training strategies (`FedABC-GL` and `FedABC-LG`). Extensive experiments demonstrate that both our approaches, `FedABC-GL` and `FedABC-LG`, outperform the state-of-the-art heterogeneous federated learning algorithms in terms of accuracy and convergence speed.

## 2. Related Work

### 2.1. Fine-tuning

The first is fine-tuning. Fine-tuning is a popular approach to adapting large-scale models to local task-specific data sets. `GTr` trained by the FL strategy requires a large amount of data, many local clients, and solid computing engines. Small companies or individuals may not be able to afford the high training cost. An economical way to utilize large models for specific usages is to fine-tune the federated pre-trained model on a local dataset. For example, [9] develop faster and sparser algorithms to fine-tune large-scale pre-trained language models differentially privately. It has been shown to enable device personalization [7] and freeze-base fine-tuning is a strategy to apply a large-scale pre-trained model to lightweight mobile devices by freezing the base layers and fine-tuning only the top layers [8]. Local-rank fine-tuning is a technique for discovering the low-rank structure in large-scale federated models and fine-tuning the low-rank part in the local client, which is widely used for fine-tuning large language models (LLMs), for example, the GPT series [9–11, 29]. For fine-tuning LLMs, LoRA [30] shed light on parameter-efficient fine-tuning in the low-rank subspace for transformer blocks. The variants of LoRA [31–35] extend low-rank adaption techniques to efficiently adapt large models to devices with limited capacity [36] However, a potential problem with fine-tuning is that the later stage `LTr` may erase or forget the information from the first stage `GTr` training [12]. *To address this, our proposed method interlaces the functions of `GTr` and `LTr` to alleviate the degradation of effectiveness in fine-tuning.*

### 2.2. Variance Reduction

The second way to improve performance in Hetero-FL scenario is to reduce the variance among clients use variance reduction techniques. SVRG [37, 38], SAGA [39], SCSG [40], and SARAH [41] are the main approaches to reducing variance for convex and non-convex optimization in a centralized fashion. FedAdpt [13], Scaffold [14] and VRLSGD [15] employ the variance reduction technique in

the distributed scenario. Cluster [16] groups the client population into groups when the FL reaches a stationary point. FedProx [6] adds a proximal term to the loss function to bring the local gradients back to the global one. SCAFFOLD [14] introduces control variates, known as gradient corrections, to reduce the variance of client updates. CANITA [42] provides gradient compressed methods for a convex setting, while MARINA [43] for non-convex ones. CONFIG and FRECON [44], incorporates communication compression with a reduction in client variance to alleviate the large number of heterogeneous clients in FL. However, these methods require an estimate of the gradient with full samples to correct the local client's gradient, which exerts a large computational complexity. *To address this, our method exaggerates the importance of `LTr` and motivates the `GTr` to find better parameters.*

### 2.3. Local Adapter

Personalized Federated Learning is a technique for quickly adapting deep neural networks for a variety of applications, such as acoustic models [45] and visual models [46]. Federated multitask learning [18] uses a mapping matrix to model the relationship between different tasks, in a convex environment. VIRTUAL [19] uses the Bayesian network to parameterize the central server and all local clients, and the variational methods to perform inference. FedEM [20] is a federated algorithm similar to expectation maximization (EM) that requires the global trainer to learn M shared complementary models, and each client learns its personalized linear weights. Ditto [21] updates the gradient with a two-step local optimization, one for the global objective function and one for the local global regularized objective function. Model-agnostic learning [22] attempts to learn global and local information simultaneously, but does not perform enough local operations to capture fine-grained features. In conclusion, in the most personalized FL framework, `GTr` and `Local` work together, but ignore the essential adversarial relationship, which is necessary to achieve better accuracy. *To address this issue, we develop an interlacing scheme with a static exchange interval to enhance both the `GTr` and `Local` functions.*

## 3. Methodology

### 3.1. Problem

The optimization in a standard FL is described in the following. Suppose that we have $m$ clients, the client $i$ holds data set $\mathcal{D}_i$. Due to data security and privacy regulations, the client $i$ may not share its private data outside the domain. For each client $i$, the local loss function can be defined by its local data set $\mathcal{D}_i$ and its local task.

$$f_i(w_i) = 1/N_i \sum_{(x_{i,j}, y_{i,j}) \in \mathcal{D}_i} f_{i,j}(w_i) = 1/N_i \sum_{(x_{i,j}, y_{i,j}) \in \mathcal{D}_i} \ell\left(x_{i,j}, y_{i,j}, w_i\right), \tag{1}$$

where $(x_{i,j}, y_{i,j})$ is the data pair in the local data set, $N_i$ is cardinality of local dataset $\mathcal{D}_i$, $w_i$ is the parameterization of the local model, $\ell$ is the objective function defined by the local task. The global loss function is defined by

$$f(w) = \sum_{i \in [m]} \alpha_i f_i(w), \tag{2}$$

where $w$ is the global model parameters, $f_i$ is the local loss function defined by Equation (1), $\alpha_i$ is the importance weights of the local loss function. Therefore, the local optimization problem is to minimize the local loss function in each client $\text{argmin}_{w_i} f_i(w_i)$ and the global optimization problem is to minimize the global loss function, i.e., $\text{argmin}_w f(w)$. There is an inconsistency between the two optimization problems due to the different data scale.

### 3.2. Motivation

In standard FL training, local data are used to train the global model on each client to ensure that the model parameters are suitable for each data set. An average operation is then used to collect

---

**Algorithm 1** Federated **A**dversary **B**ased **C**ooperation (`FedABC`)

---

**Input**: $w_i^0$ initialization of local model; $\mathcal{D}_i = \{(x_{i,j}, y_{i,j})\}$ data set holds by Client;
**Hyper-parameter**: $\alpha_i$ importance of local clients; $\beta_1^t$ global learning rate; $\beta_2^t$ local learning rate; $B$ batch size; $E_1$ global update interval; $E_2$ local update interval; $T$ total iteration steps.
**Output**: $w^T$ model parameter at final step.

1: Alternatively update models with `GTr` and `LTr` Stages;
2: **Global Trainer (`GTr`)**
3: Client $i$ samples a batch of data $\mathcal{B}_i$ of batchsize $B$;
4: **while** $t \in E_1$ **do**
5:     Compute gradient $g_i^t = 1/|\mathcal{B}_i| \sum_{j \in \mathcal{B}_i} \nabla f_{i,j}(w_i^t)$;
6:     Clients send local gradients $g_i^t$ to Server;
7:     Server aggregate the local gradients $g_{\text{global}} = \sum_{i \in [m]} \alpha_i g_i$;
8:     Update global parameter $w^{t+1} = w^t - \beta_1^t g_{\text{global}}^t$;
9:     Server distributes global parameters $w^{t+1}$ to Clients;
10:     Client update local parameters $w_i^{t+1} \leftarrow w^{t+1}$;
11: **end while**
12: **Local Trainer (`LTr`)**
13: **while** $t \in E_2$ **do**
14:     Client $i$ samples a batch of data $\mathcal{B}_i$ of batchsize $B$;
15:     Compute gradient $g_{\text{local}}^t = 1/|\mathcal{B}_i| \sum_{j \in \mathcal{B}_i} \nabla f_{i,j}(w_i^t)$;
16:     Update local model parameter $w_i^{t+1} = w_i^t - \beta_2^t g_{\text{local}}^t$;
17: **end while**
18: **return** the trained global model parameter $w^T$.

---

the knowledge from different clients to serve the global model. This procedure implies that the global parameters obtained differ from those obtained locally and that the fine-grained information obtained on each client may be lost during the average process. These local processes bring to the systems a comprehensive knowledge of data from various clients. As the global training process progresses, the gains of each epoch decrease and the locally obtained fine-grained information, which the global operation may lose, accumulates. Therefore, additional local processes are needed to find them. Quite surprisingly, local processes may not help the global training process. Our method proposes an alternate training process as G-L-G-L, while each stage, `GTr` or `LTr`, consists of several global or local training epochs. Increasingly lost fine-grained information is learned and accumulated through a local training process. Unlike existing strategies, `GTr` and `LTr` updates co-occur. Our methods process a static interval exchange between `GTr` and `LTr`. We design the interval exchange strategy because the training for `GTr` and `LTr` needs several epochs to exert its influence on overall performance. Learning both `GTr` and `LTr` in one step would omit the adversarial relation.

### 3.3. Solution

To resolve the inconsistency in the optimizations of `LTr` and `GTr`, we propose the following algorithm `FedABC`. We describe the algorithm of our proposed Federated **A**dversary-**B**ased **C**ooperation **G**lobal **L**ocal (`FedABC-GL`) in Algorithm 1. We alternate between **Global Trainer (`GTr`)** and **Local Trainer (`LTr`)** during each update stage. For each sample $j$ of the client $i$, the sample-wise gradient at time step $t$ is $g_{i,j}^t = \nabla f_{i,j}(w_i^t)$. For client $i$, the client-wise gradient at time step $t$ is $g_i^t = 1/|\mathcal{B}_i| \sum_{j \in \mathcal{B}_i} g_{i,j}$. For `GTr`, the gradients are updated by aggregating the gradients of local clients, that is, $g_{\text{global}} = \sum_{i \in [m]} \alpha_i g_i$. For `LTr`, it updates the model parameters with one single gradient of client $i$, $g_{\text{local}} = g_i$. Let $E_1$ denote the training epochs for `GTr`, while $E_2$ denote the training epochs for `LTr`. Then we have the alternative update formula as

$$
\begin{aligned}
w^{t+1} &= w^t - \beta_1^t g_{\text{global}}, \quad t \in E_1; & (3)\\
w^{t+1} &= w^t - \beta_2^t g_{\text{local}}, \quad t \in E_2. & (4)
\end{aligned}
$$

Algorithm 1 outlines the comprehensive procedures for `GTr` in Lines 4-11 and for `LTr` in Lines 13-17.

Table 1: Overall averaged test accuracy with eleven methods and four data sets for HFL tasks.

| Datasets | FedAvg | FedAdpt | Cluster | Local | FedProx | FedEM | AFL | Finetune-30 | Finetune-60 | FedABC-GL | FedABC-LG |
|---|---|---|---|---|---|---|---|---|---|---|---|
| **CIFAR10** | 80.85 | 81.64 | 77.04 | 80.69 | 72.84 | 85.81 | 76.80 | 84.13 | 84.46 | **90.12** | 89.45 |
| **CIFAR100** | 53.62 | 45.70 | 53.37 | 51.53 | 45.02 | 53.88 | 47.88 | 57.40 | 58.32 | **67.39** | 67.18 |
| **EMNIST** | 82.97 | 85.31 | 82.90 | 83.21 | 81.61 | 84.66 | 83.20 | 85.68 | 86.52 | **88.18** | 87.96 |
| **FEMNIST** | 80.05 | 51.12 | 67.68 | 67.01 | 56.75 | 76.53 | 81.02 | 60.25 | 70.45 | **81.16** | 76.12 |

Due to the duality of the two approaches, they share similar performance in most scenarios with an interlacing training tendency. Furthermore, we analyze the roles of `GTr` and `LTr` in the training process of `FedABC-GL` and `FedABC-LG`. We observe consistent behaviors of `GTr` and `LTr` where `GTr` improves the global learning accuracy and decreases the global loss, and `LTr` contributes to increasing the global loss while pushing the model parameter to a better restart position. The multiscale concept in Algorithm 1 allows the selection of $\beta_1$ and $\beta_2$ at different scales, enabling the `GTr` to capture global information and `LTr` to obtain local ones.

# 4. Evaluation

In this section, we perform comprehensive experiments to evaluate the model in order to address the following essential research questions.

- **RQ1.** How effective are our methods (`FedABC`) compared to existing state-of-the-art hierarchical federated learning methods (SOTA)?

- **RQ2.** What are the roles of `GTr` and `LTr`, and how does the transition strategy contribute to federated training?

- **RQ3.** What is the long-term behavior of the proposed methods compared to the existing ones?

## 4.1. Experimental Setup

Initially, we describe the experimental parameters by outlining the configurations for five data sets and eight comparative methods. In addition, we discuss the specifics of the model architectures and hyperparameters for each experiment group.

### 4.1.1. Datasets

- **CIFAR10**: Contains 60,000 images in ten classes (6,000 training images and 1,000 test images per class), each at 32x32 pixel resolution. Divided among ten clients.

- **CIFAR100**: Similar to CIFAR10 but with 100 classes, each having 600 training images and 100 test images. Also divided among ten clients.

- **EMNIST**: Comprises 1,120,000 training images and 560,000 handwritten digit test images in 62 classes, at a resolution of 28x28 pixels. Divided among 100 clients.

- **FEMNIST**: Includes 260,000 training images and 87,000 test images of handwritten letters from 10 classes, at a resolution of 28x28 pixels. Divided among 500 clients.

### 4.1.2. Baseline

Here we present the details of the eight methods compared. `FedAvg` [1] is the de facto method in federated learning. Train multiple clients simultaneously and average the model parameters every few steps, which can be viewed as a global trainer in our design. FedAdpt[13] is `FedAvg` with local tuning. Cluster[16] group the client population into groups when FL reaches a stationary point. FedProx[6] is `FedAvg` with a proximal term as the objective function. FedEM[20] is an EM-like federated algorithm where the global trainer learns M shared complementary models and each client learns its personalized linear weights. AFL[47] is an abbreviation for agnostic federated learning, in which the objective of the global model is to approximate any target distribution that is made up

of the distributions of local clients. Local is the isolated training paradigm in which each client is trained solely on its local dataset, which can be seen as a local trainer in our design. Finetune-30 and Finetune-60 are the two-stage training paradigm with Global trainer (`FedAvg`) in the first 30 epochs and 60 epochs, respectively.

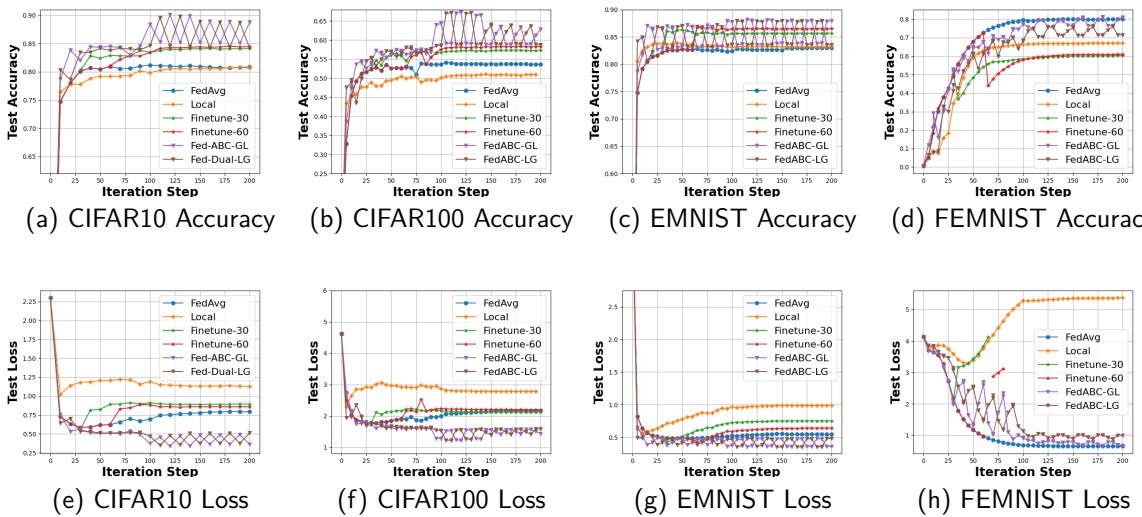

Figure 2: The overall convergence of `FedABC-GL` and `FedABC-LG` compared with selected existing FL methods.

### 4.1.3. Model Architectures

The purpose of this study is to explore the roles of global and local training in FL and to devise an update formula that is more effective than simply combining the two. Our primary concern is not the creation of a new model structure. Therefore, we use the same model configurations as those used in the previous study [20]. For CIFAR10 and CIFAR100, we utilize a pre-trained mobilenet-v2 with cross-entropy serving as the loss function. The sole distinction between the two data sets is the class count, with CIFAR10 having 10 and CIFAR100 having 100. For EMNIST and FEMNIST, we implement a two-layer CNN. For SHAKESPEARE, we employ a single-layer LSTM model to predict the next word.

### 4.2. Overall Performance Improvement (RQ1)

In this section, we demonstrate the overall performance of `FedABC` and compare it with all baselines using the metric 'Average Test Accuracy' as shown in Table 1. Compared to existing federated learning algorithms, including `GTr` (`FedAvg`), `Local`, fine-tuning methods (Finetune-30 and Finetune-60) and personalized FL techniques (FedAdpt, FedEM, AFL, Cluster, and FedProx). More concretely, we evaluated the final average test accuracy over five repetitions for all ten methods trained on four different heterogeneously split datasets. To establish the superiority of our approaches over the leading baselines, we perform a significance test where a p-value $< 0.05$ indicates a statistically significant improvement by `FedABC`. From the experimental results presented in Table 1, we obtain the following insight. Our methods (`FedABC`) significantly outperform nearly all baseline models, including strong heterogeneous Federated Learning (FL) methods such as FedApt, AFL, and FedEM. Notably, we improved the best test accuracy from 80.95% to 90.12% for CIFAR10 and from 53.62% to 67.92% for CIFAR100. For EMNIST and FEMNIST, we observed an improvement in accuracy of 6% compared to `FedAvg`, which we attribute to the interchanging training phase. Furthermore, a better accuracy of the local paradigm leads to greater improvements with our proposed methods (`FedABC`and `FedABC-LG`), thanks to the contributions of the local trainer.

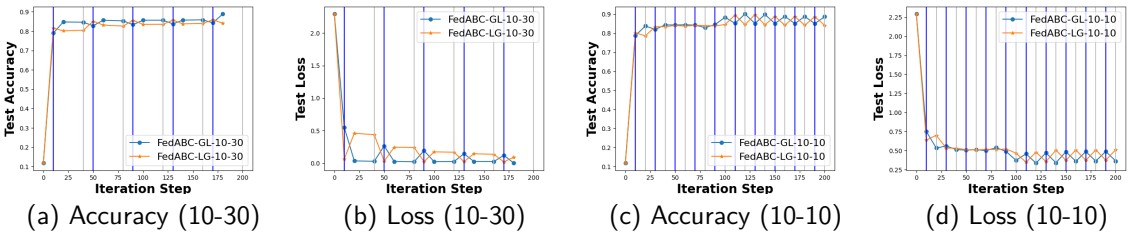

| (a) Accuracy (10-30) | (b) Loss (10-30) | (c) Accuracy (10-10) | (d) Loss (10-10) |

Figure 3: Transition between `LTr` and `GTr`. The blue vertical line denotes one transition point, while the silver vertical line denotes another transition point.

## 4.3. GL Strategy Struggles by Oscillation (RQ2)

### 4.3.1. Overfiting phenomenon

In this section, we investigate the effect of alternating between `GTr` and `LTr` on the transition in the training landscape. From the experimental results in the selected datasets, we observe that `LTr` (yellow line) tends to overfit at an early stage; for example, the yellow line begins to increase at Epoch 10 with the **CIFAR10** data set and at Epoch 40 with the FEMNIST data set. This observation implies that `LTr` fits a local minimum in the training landscape, but cannot perform better in the test landscape. On the other hand, `GTr` (blue line) tends to overfit later; for example, the blue line begins to increase at epoch 80 with the CIFAR10 data set and does not increase with the FEMNIST data set. The fine-tuning trainers (green and red lines) behave between `LTr` and `GTr`, i.e., they decrease in the first training stage and start to increase when changing from `GTr` to `LTr`. Our proposed methods `FedABC-GL` (brown) and `FedABC-LG` (purple) keep decreasing until the alternating patterns appear. Our methods achieve the smallest test loss throughout the oscillation stages among all methods.

### 4.3.2. Transition of `LTr` and `GTr`

Next, we present a detailed analysis of the transitions between the `LTr` and `GTr`. We present the transition points with the test accuracy and the tendency for loss in Figure 3. One can observe the roles of `LTr` and `GTr` from Figure 3. Specifically, for `FedABC-GL` (light blue curve), the blue vertical line denotes the transition from `GTr` to `LTr` while the silver vertical line denotes the transition from `LTr` to `GTr`. And the opposite for `FedABC-LG` (orange curve). When translates from `GTr` to `LTr`, the accuracy increases while the loss decreases for `FedABC-GL` and `FedABC-LG`. However, for translation from `LTr` to `GTr`, the accuracy decreases, while the loss increases. This phenomenon indicates that `GTr` cannot spontaneously obtain the fine-grained information grasped by the output of `LTr`. Furthermore, the orange curve with the G-10-L-30 stage tends to be flat at intervals of length 30, which means the long term training with `LTr` although does not contribute to `GTr` metric but keep the parameters in good position near it. For stage G-10-L-10, we observe that both `FedABC-GL` and `FedABC-LG` struggle to find the best pattern and achieve the best accuracy as long as they arrive at the alternating pattern.

### 4.3.3. Why the oscillation occurs and its benefits

The main motivation for `FedABC` is that neither `GTr` nor `LTr` training can be sufficient for all users. Traditional personalized approaches try to find a balance between global and local trainers but do not solve the problem. Training in a global fashion still leads to errors due to differences between users, while training in a local fashion is hindered by the generalization error caused by the limited amount of local data. The oscillation arises from the varying levels of information that `GTr` and `LTr` gather during the training phase. `GTr` is attuned to the common attributes across all clients, whereas `LTr` is focused on the distinct traits of an individual client. This variance in the descent direction induces an oscillation in the iterative alternating process. An adversary can foster successful collaboration by

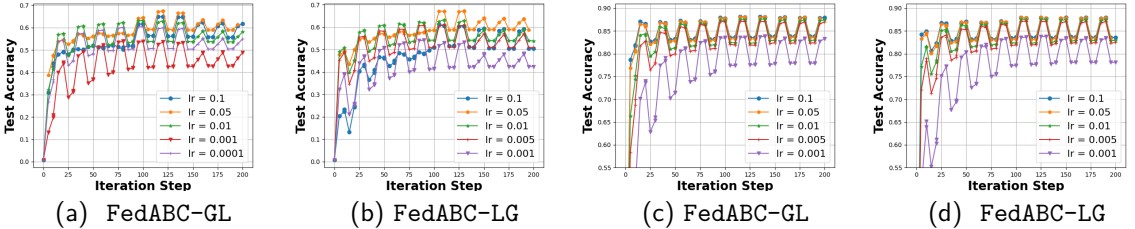

Figure 4: The influence of learning rates on CIFAR100 (left two images) and EMNIST (right two images).

preventing a single trainer from getting trapped in a local minimum and simultaneously prompting the two trainers to delve deeper into the data set. This is the reason why oscillation is believed to enhance performance.

## 4.4. Long Term Behavior (RQ3)

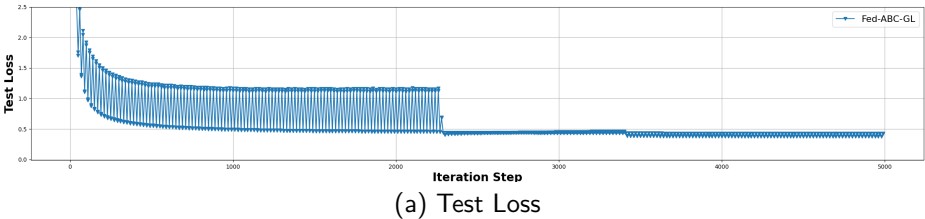

(a) Test Loss

Figure 5: The long term test accuracy on FEMNIST training with `FedABC-GL`.

In this section, we examine the prolonged behavior of `FedABC` training. As stated earlier, `LTr` focuses on moving towards the optimal point for each individual, while `GTr` aims to enhance the average performance. By alternating between `GTr` and `LTr`, we notice an upward trend in the learning curve, which results in better performance compared to `GTr`, `LTr`, and more personalized approaches. We now investigate the long-term behavior for `FedABC`. To do this, we use an example on FEMNIST with 5000 training iterations and present the long-term convergence in Figure 5. `LTr` and `GTr` display an increasing tendency and potential to jump, while `FedABC-GL` accelerates jumping in the early stage of training and continues to oscillate between a high accuracy point and a low accuracy point after jump. In particular, we also observe `FedABC-GL` to double ascent accuracy in long-term behavior.

## 5. Conclusion

This article introduces a new training paradigm, named `FedABC`, which combines `GTr` and `LTr` to take advantage of both and reduce the drawbacks of each. We performed a special analysis of adversary-based cooperation of `GTr` and `LTr`, which has been neglected in previous studies. Specifically, `FedABC` alternates between `GTr` and `LTr` at predetermined intervals. Several consecutive local training steps reveal more detailed information and guide the global process to a new pattern that cannot be generated spontaneously by `GTr` alone. We propose two dual approaches, `FedABC-GL` and `FedABC-LG`, depending on whether `GTr` or `LTr` is used first. Comprehensive experiments show that `FedABC` achieves higher accuracy, faster convergence compared to the existing SOTA. Further research will explore a rigorous and quantitative analysis of the convergence behavior and training dynamics of `FedABC`.

# Acknowledgements

This work is supported by the **National Natural Science Foundation of China** (No. NSFC92370205, 12271512), the **Zhejiang Province Key Research and Development Plan** (No. 2024SSYS0010), and the **National Key Research and Development Program of China** (No. 2023YFB2703700).

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

## A. Hyper-parameters and Implementation Details

For a fair comparison, we follow the same hyperparameters as in the previous work [13, 16, 20]. For all datasets, we set the learning rate at $\eta$ in $\{0.001, 0.005, 0.01, 0.03, 0.05, 0.1\}$ and select the best performance among the different learning rates to report overall performance. We also check the influence of learning rates, especially for our methods. We set the batch size at 128 and the iteration rounds at 200. For the FedEM [20] method, the number of mixture models is set to 3. Hsu [48] uses Dirichlet sampling with a hyperparameter $\alpha$ to define the heterogeneous distribution of the data and proposes a label-based Dirichlet partition method. An increase in $\alpha$ corresponds to a more significant heterogeneity. In our experiments, all data sets were divided by $\alpha = 0.4$. Our experimental evaluations are conducted on a computational platform equipped with four NVIDIA Tesla A100 GPUs. Each experiment is conducted five times, and the mean accuracy is displayed.

## B. Long Term Behavior

In this section, we analyze the long-term behavior of the proposed methods in more data sets. We present the test accuracy and loss for `GTr` (FedAvg), `LTr` (Local), and `FedABC-GL` in Figure 6 respectively. First, we observe an increasing trend for all three methods in test accuracy. However, for test loss, `LTr` tends to first decrease and then increase in both the CIFAR10 test loss and the EMNIST test loss (Figure 6 (b) and (d)). This is the verfication that `LTr` solely shows a bad performance on the generalization in general due to the small amount of training data it used. This is also the reason that hinders `LTr` achieves comparable accuracy with the `GTr`  For CIFAR10, `LTr` jumps around 2500 iteration steps, `GTr` gradually improves after 200 iterations, and `FedABC-GL` jumps around 110 iterations and fluctuates between 0.85 and 0.90 afterward. For EMNIST, all three methods show an increasing trend in the early stage (before 200 epochs). After that, the test accuracy of `LTr` and `GTr` remain around 0.835 and 0.823, respectively. In contrast, `FedABC-GL` quickly converges to 0.87 around 200 iteration steps and then varies between 0.83 and 0.87. After around 2500 iteration steps, the green curve experiences a slight jump to around 0.875 (accuracy). We also observe the **double ascent phenomenon** for test accuracy, while the **double decent** for test loss on CIFAR10 and EMNIST.

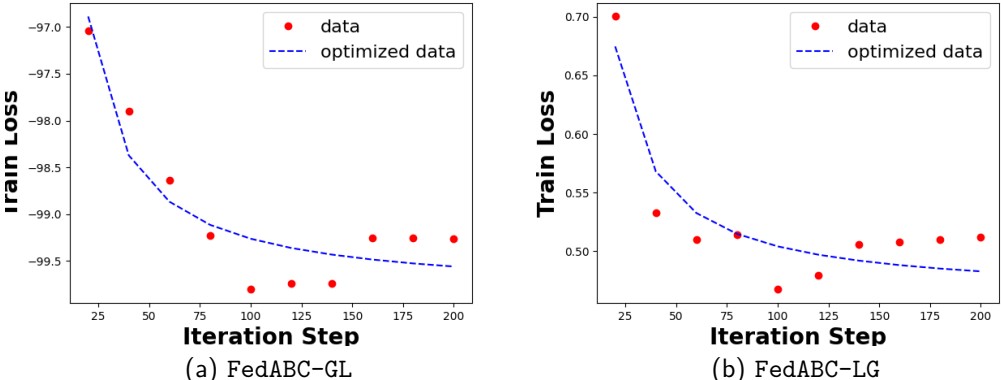

(a) FedABC-GL  (b) FedABC-LG

Figure 7: The fitting curve for train loss on CIFAR10.

## C. Influence of Learning Rates

We investigate the influence of learning rates on `FedABC-GL` and `FedABC-LG` with three datasets. Table 2 shows the highest test accuracy achieved with different learning rates. Generally, higher learning rates lead to better convergence for most datasets. For instance, `FedABC-GL` obtained 67.39% accuracy on CIFAR100 with $\eta = 0.05$ and 88. 18%, 81. 07% in EMNIST and FEMNIST

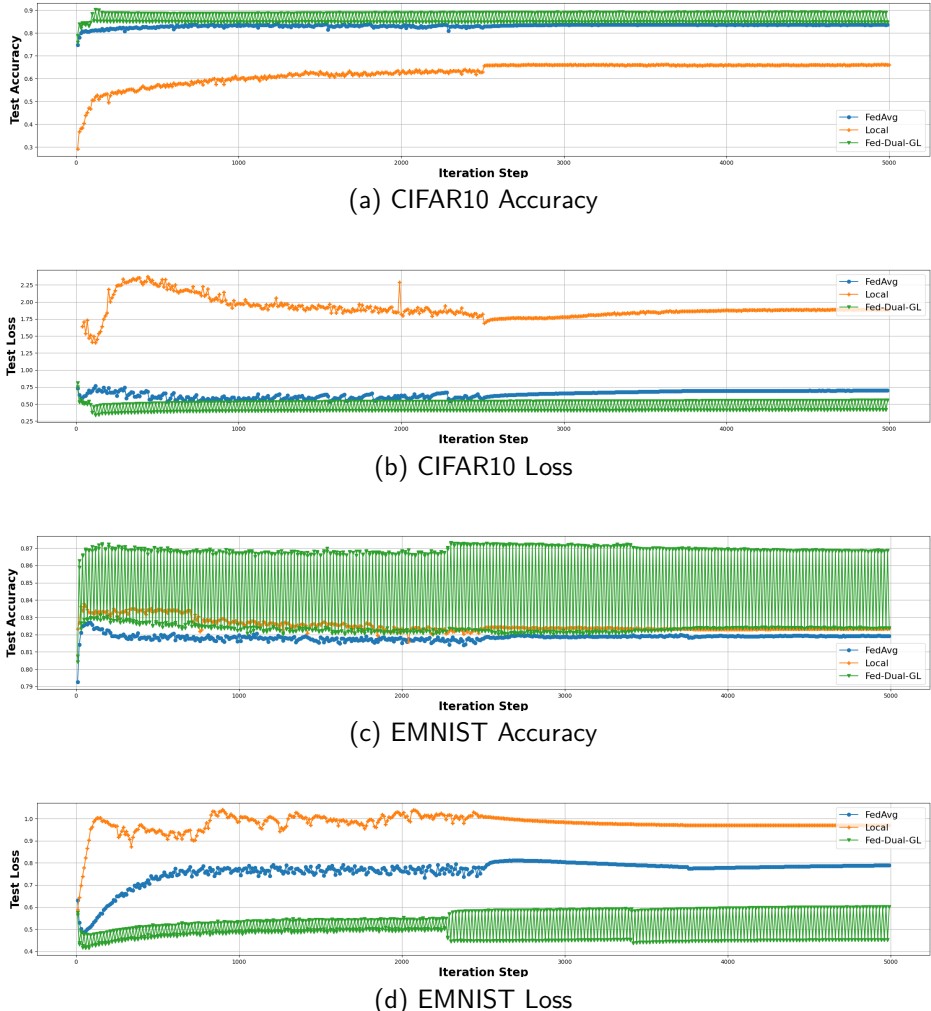

(a) CIFAR10 Accuracy

(b) CIFAR10 Loss

(c) EMNIST Accuracy

(d) EMNIST Loss

Figure 6: The long term test accuracy and loss on CIFAR10 and EMNIST.

with $\eta = 0.1$. `FedABC-LG` performed best on EMNIST with $\eta = 0.1$ and CIFAR10, FEMNIST with $\eta = 0.05$. The test accuracy follows a roughly anti-U shape as the learning rate increases; it starts by increasing and then decreases. More importantly, the range of accuracy influenced by learning rates is large. By tuning the best learning rates, one can improve the accuracy of CIFAR100 up to 14.16 %. And for FEMNIST, we have a more significant improvement of up to 49. 08%.

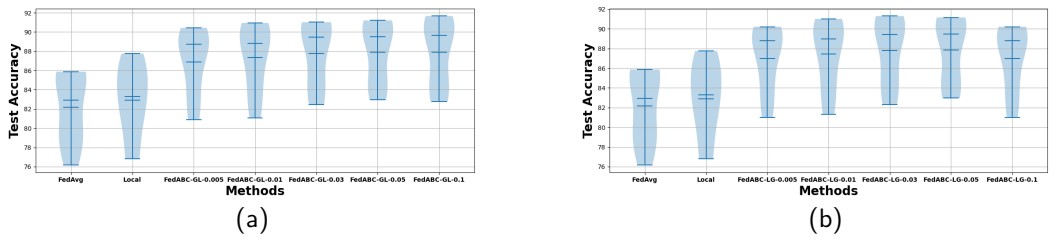

(a)                                        (b)

Figure 8: Violin Plots of Test Accuracy among 10 Clients on **EMNIST**. The upper graph shows the comparisons of `FedABC-GL` with `GTr` trainer (`FedAvg`) and `LTr` (`Local`) trainer while the lower graph shows the comparisons of `FedABC-LG`.

Table 2: Influence on Learning Rates of Different Dataset

| Datasets | FedABC-GL | | | | | |
|---|---|---|---|---|---|---|
| | GL-0.0001 | GL-0.001 | GL-0.01 | GL-0.03 | GL-0.05 | GL-0.1 |
| CIFAR100 | 60.45 | 53.23 | 58.11 | 65.72 | **67.39** | 61.73 |
| EMNIST | 87.11 | 83.79 | 87.60 | 87.96 | 88.06 | **88.18** |
| FEMNIST | 31.99 | 11.46 | 44.06 | 70.14 | 76.04 | **81.07** |
| | FedABC-LG | | | | | |
| | LG-0.0001 | LG-0.001 | LG-0.01 | LG-0.03 | LG-0.05 | LG-0.1 |
| CIFAR100 | 57.11 | 54.27 | 62.98 | 66.48 | **67.18** | 59.18 |
| EMNIST | 87.15 | 83.77 | 87.55 | 87.91 | 87.92 | **87.96** |
| FEMNIST | 30.44 | 10.54 | 44.42 | 70.18 | **76.12** | 73.34 |

## D. Convergence Speed Fitting

We investigated the convergence speed of `FedABC-GL` by fitting the train loss on the **CIFAR10** dataset. We use the function $f(x) = x/a + b$ to fit, where the values of $a$ and $b$ are 59.41 and $-99, 85$, respectively. The red dots in Figure 7 represent the points we selected from the train loss of `FedABC-GL` with learning rate $\eta = 0.05$. The blue curves correspond to the values taken from the function $f(x) = x/59.41 - 99.85$. We omit the upward points and only consider the downward points because of the oscillating nature. The reason why we use the train loss to fit the convergence rate is that it accurately reflects the optimization procedure of the objective function, while the test loss contains the generalization error term. The fitting results verify that our methods show a convergence with a decay rate of $O(1/n)$, while $n$ is the number of iteration steps experimentally. For `FedABC-LG` , we observe similar fitting results, while the fitting function is $f(x) = x/4.26 + 0.46$.

## E. Fairness Improvement

In addition, we show the improvement in fairness of our techniques in comparison to `FedAvg` and the `Local` using a violin plot (Figure 8). The two left columns illustrate the accuracy distribution for both `FedAvg` and `Local`, whereas the final five columns depict the accuracy distribution for `FedABC-GL` and `FedABC-LG`, across a range of learning rates, including $\{0.005, 0.01, 0.03, 0.05, 0.1\}$. Upon examining Figure 8, it is evident that both the proposed methods including `FedABC-GL` and `FedABC-LG` enhance the average accuracy among clients and decrease the standard deviation when compared to `FedAvg` and `Local`. `FedABC-GL` demonstrates a steady enhancement in the distribution of the mean accuracy between clients. `FedABC-LG` displays a nearly consistent improvement in the same distribution; however, it exhibits a slight decrease in the mean accuracy when the learning rate is elevated (for example, $\eta = 0.1$). This observation suggests that an increased learning rate could lead to greater oscillations. Lastly, the `Local` method exhibits the narrowest difference between the mean and median value of the accuracy distribution, indicating a less important impact of outliers or extreme values on the comprehensive measure of the central tendency.

## F. Large-scale Interlacing on Shakespeare

We can draw several conclusions from Figure 9. First, transitions have a varying impact on test and training. When we switch from `GTr` to `LTr`, the training accuracy increases, but the test accuracy decreases. In contrast, when we move from `LTr` to `GTr`, we see an improvement in the test accuracy and a reduction in the training accuracy. Second, there is a direct relationship between loss and accuracy. As the test loss decreases, the test accuracy increases. Third, there is a significant discrepancy between training accuracy and test accuracy. Training accuracy can reach up to 0.95, while test accuracy is only 0.45. This indicates an unavoidable generalization error. Lastly, learning rates play a crucial role in the trainer's rapid pattern recognition. A higher learning rate ($\eta = 0.1$, represented by the blue line) allows the algorithm to identify the pattern more quickly. A lower learning rate ($\eta = 0.01$, represented by the red line) results in slower pattern recognition, with the

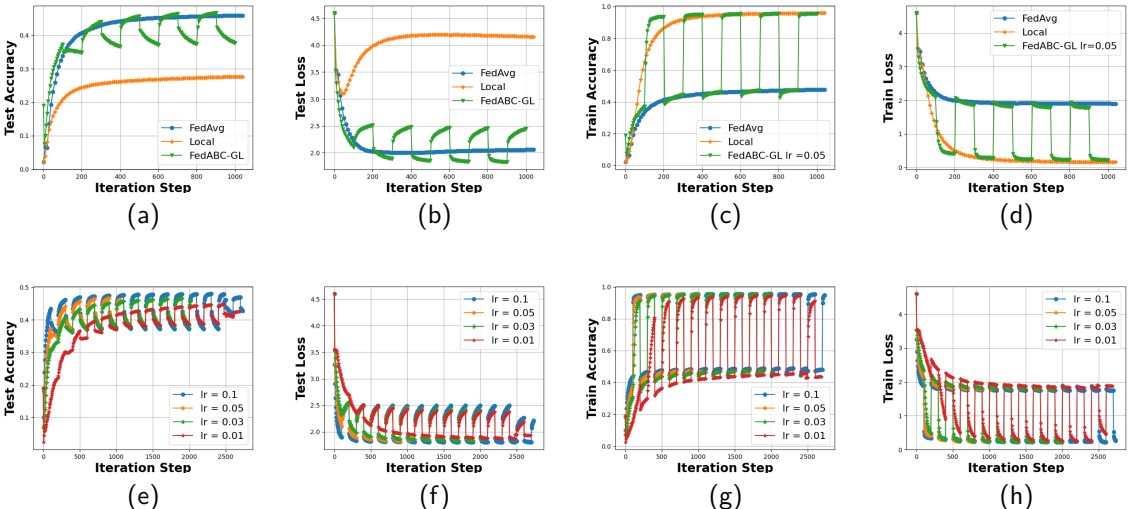

Figure 9: The first row presents a comparison between `FedABC-GL`, `FedAvg`, and `Local`. In the second row, a comparison of `FedABC-GL`with varying learning rates is depicted. The first column illustrates the accuracy of the test. The second column represents the test loss. The train accuracy is plotted in the third column, while the train loss is shown in the final column.

Table 3: A Comparison of C-Eval

| Methods | Stem | Social Sciences | Humanities | Others | Average | Avg(hard) |
|---|---|---|---|---|---|---|
| **Cent-LlaMa** | 24.5 | 25.6 | 25.5 | 24.4 | 24.9 | 23.4 |
| **FedABC-GL-LlaMa** (ours) | **25.8** | 25.7 | 25.5 | **26.3** | **25.8** | **24.5** |
| **FedABC-LG-LlaMa** (ours) | 24.5 | **27.3** | **25.7** | 25.1 | 25.4 | 21.9 |
| **FedProx-LlaMa** | 23.9 | 25.6 | 25.3 | 23.4 | 24.4 | 22.1 |
| **FedAvg-LlaMa** | 22.9 | 22.5 | 23.9 | 23.9 | 23.2 | 21.6 |
| **Base-LlaMa** | 21.6 | 23.4 | 23.9 | 23.3 | 22.8 | 20.3 |

search continuing through the initial 500 epochs before finally identifying the patterns. Once the trainer identifies the patterns, the learning rate has minimal impact on the final results. This finding aligns with the results from other vision datasets that we discuss in the main context.

# G. An Extension to the Large-language Models

This section focuses on the role of `FedABC` in federated fine-tuning of large language models (LLMs), specifically addressing the challenges posed by limited local data and privacy restrictions. Generative LLMs fine-tuned with instructions demonstrate strong generalization abilities, but performance can suffer due to statistical heterogeneity among clients. To address this, we implement `FedABC` using LlaMa [49] and a parameter-efficient fine-tuning (PEFT) strategy with low-rank adaptation (LoRA [30]). In each communication round, clients decompose the model's attention layer into low-rank matrices and send them to the server, which aggregates and returns the updated layers for local model adjustments.

We perform an experimental evaluation among the base model (Base-LlaMa), the `FedAvg` fine-tuned variant (FedAvg-LlaMa), the FedProx fine-tuned variant (FedProx-LlaMa), the `FedABC-LG` fine-tuned variant (FedABC-LG-LlaMa), the `FedABC-GL` fine-tuned variant (FedABC-GL-LlaMa), and the centralized fine-tuned variant (Cent-LlaMa). All evaluations comply with the Chinese multilevel multidiscipline evaluation suite for foundation models (C-eval) [50], which structures all questions with four options. The questions cover a wide spectrum of 52 different fields, extending from the

humanities to the realms of science and engineering. We produce answers using either the base models or the fine-tuned models and subsequently submit these answers to the online C-eval evaluation system to receive a score. The evaluation results presented in Table 3 demonstrate the benefits of our methods (`FedABC-GL` and `FedABC-LG`) compared to the three fine-tuned approaches and the foundation model. We utilize bold text to highlight the highest score for each subject and underlined text to indicate the second highest score for each subject. The C-Eval results indicate that all fine-tuned models outperform the base model, with FedAvg-LlaMa improving five subjects and FedProx-LlaMa enhancing the 'Social Sciences' score to 25.6. The centrally fine-tuned model (Cent-LlaMa) consistently exceeds the performance of federated models, achieving scores of 24.5 in 'Stem' and 23.4 in 'Avg(hard)' due to its centralized training approach. Our methods, `FedABC-GL` and `FedABC-LG`, advance Cent-LlaMa by an average of 2 points, with `FedABC-GL` leading in several categories while `FedABC-LG` notably scores 27.3 in 'Social Sciences.' Furthermore, `FedABC-GL` outperforms `FedABC-LG` in specific areas such as "plant protection" and "education science." To conclude, the models fine-tuned by our methods exhibit robust problem-solving capabilities for most subjects in the LLM application.

## H. Detailed C-Eval Results

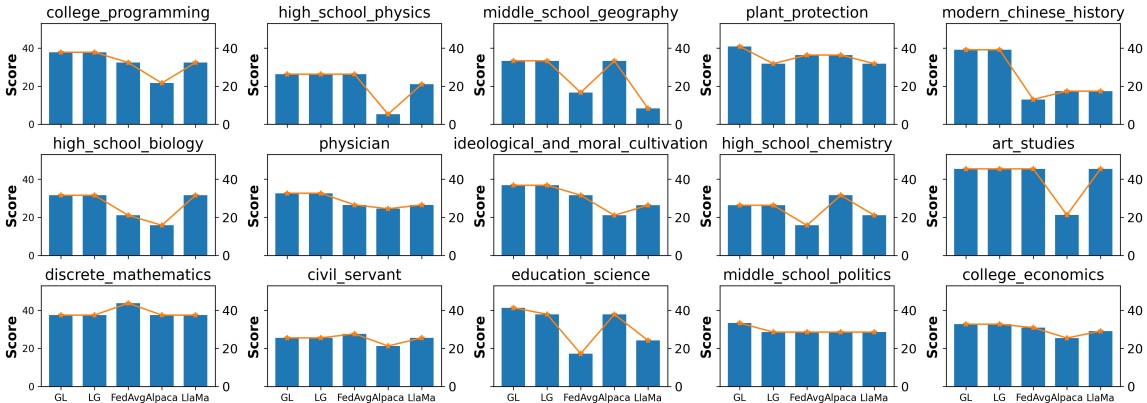

Figure 10: C-evaluation of Large Language Model Fine-tuning

For a more detailed comparison, we selected 15 specific subjects for evaluation and present the scores in Figure 10. Alpaca is another foundation model, refined from LlaMa [51]. A consistent conclusion can be drawn from the results summarized in Figure 10. To be specific, all test cases benefit from fine-tuning, as evidenced by the improved scores of the fine-tuned models (`FedABC-GL`, `FedABC-LG`, and `FedAvg`) compared to the base models (LlaMa and Alpaca). Secondly, `FedAvg` outperforms in assessments related to 'discrete mathematics' and 'civil servant', resulting in the bell-shaped curve. Thirdly, `FedABC-GL` and `FedABC-LG` secure the highest scores in most scenarios, including "modern Chinese history", "ideological and moral cultivation", and "high school biology". Lastly, `FedABC-GL` outperforms its dual version `FedABC-LG` in certain contexts such as "plant protection" and "education science", aligning with the experimental results we observed in vision classification tasks (refer to Table 1). To conclude, the models fine-tuned by our methods exhibit robust problem-solving capabilities for most subjects.

