# OpenReview forum: "Do Global and Local Perform Cooperatively or Adversarially in Heterogeneous Federated Learning?"
_CPAL.cc/2025/Proceedings_Track — CPAL 2025 (Proceedings Track) Poster_

### Official Review · Reviewer_evQN · 2025-01-10
**A global and local training paradigm for federated learning**

**Rating:** 7
**Confidence:** 4

**Review:**

This paper introduces FedABC to incorporate global learning with local training of different clients for FL. The motivation and intuition behind such training paradigm is sound and reasonable. I like the observation of Oscillation occurs at the transition phase which in turn leads to better generalization looks very intriguing, which deserves more deep understanding. The performance gain over FedAvg and fine-tune demonstrates the benefits of this co-training strategy.

---

### Official Review · Reviewer_siLJ · 2025-01-13

**Rating:** 4
**Confidence:** 4

**Review:**

**Summary:**

This paper examines the relationship between global and local trainers in Federated Learning (FL) and introduces a novel cooperation approach between global and local federated optimizers, FedABC. Unlike conventional strategies that balance global and local models, FedABC leverages alternating adversarial dynamics between global and local trainers to improve overall performance. Experimental results demonstrate that the proposed method achieves better accuracy and faster convergence compared to other baseline FL methods, showcasing its efficacy in addressing statistical heterogeneity while maintaining fairness and generalizability across diverse datasets.

**Pros:**
- The paper is well-written and well-motivated overall.
- The proposed FedABC method is intuitive and easy to understand.
- The empirical results of the proposed FedABC method are promising.

**Cons:**
- A typical assumption in FL is that no data is available on the global server. If this is the case, how is it possible to execute Line 5 in Algorithm 1?
- The experiments for the proposed FedABC method are conducted on overly simple tasks, such as MNIST and CIFAR-10. It is essential to include larger-scale experiments.
- The proposed method lacks theoretical justification. Does FedABC have any theoretical guarantees of better or faster convergence compared to other federated optimizers?
- The test accuracy seems to oscillate between federated rounds (as shown in Figure 2). Is this due to instability in the training of FedABC?

---

### Official Review · Reviewer_bf3x · 2025-01-14
**Limited novelty but solid experiments**

**Rating:** 7
**Confidence:** 3

**Review:**

Summary:

The paper presents a syncoziation approach to addressing statistical heterogeneity in Federated Learning (FL). The authors introduce FedABC, an alternating training strategy based on adversarial cooperation between a Global Trainer (GTr) and a Local Trainer (LTr). This innovative method proposes a scheduling system for global and local updates.

Strengths:
- The paper provides extensive experimental results across multiple datasets (CIFAR10, CIFAR100, EMNIST, FEMNIST), and compare with many baseline methods. FedABC achieves notable improvements in accuracy (up to 13.77%) and convergence speed.
- The method is easy to follow.

Weakness:
- The paper's technical novelty is somewhat limited. While the empirical results are compelling, a more rigorous theoretical analysis of the convergence behavior and the underlying dynamics of FedABC would significantly enhance the paper's contributions.
- Section 3.3 lacks sufficient mathematical formulations to clearly illustrate the scheduling system and the interaction between the Global Trainer and Local Trainer. Adding more detailed equations and derivations would improve the clarity and technical depth of this section.

Questions:
- What does "Adversarially" refer to in the context of the proposed method?

---

### Meta-Review · Area_Chair_2gtV · 2025-02-06

**Recommendation:** Accept (Poster)
**Confidence:** 3

**Metareview:**

The paper introduces FedABC (Federated Adversarial Based Cooperation) for heterogeneous federated learning (Hetero-FL). This method utilizes an alternating training strategy between a Global Trainer (GTr) and a Local Trainer (LTr), described as an adversarial cooperation. Unlike traditional methods that primarily focus on balancing global and local information, FedABC intentionally creates a dynamic tension between the two, allowing for adversarial interactions that enhance overall model performance.

FedABC demonstrates significant improvements in accuracy and convergence speed over state-of-the-art heterogeneous FL methods through rigorous theoretical analysis and extensive experiments on various datasets. This approach not only addresses the challenge of statistical heterogeneity in the data across different clients but also ensures that both global and local perspectives are effectively utilized to enhance the learning process.

It is suggested that the author consider the reviewers' comments in their final revision.

---

### Decision · Program_Chairs · 2025-02-11

Accept (Poster)